# Efficacy of Antihypertensive Therapy in a Child with Unilateral Focal Fibromuscular Dysplasia of the Renal Artery: A Case Study and Review of Literature

**DOI:** 10.3390/medicines7020009

**Published:** 2020-02-20

**Authors:** Ratna Acharya, Savannah Ellenwood, Kiran Upadhyay

**Affiliations:** 1Department of Pediatrics, University of Florida, Gainesville, FL 32610, USA; 2Department of Pediatrics, Division of Pediatric Nephrology, University of Florida, Gainesville, FL 32610, USA

**Keywords:** Renal artery stenosis, fibromuscular dysplasia, hypertension

## Abstract

**Background**: Fibromuscular dysplasia (FMD) is one of the important etiologies of renovascular hypertension in children. It is usually resistant to multiple antihypertensive agents and can cause extreme elevation in blood pressures, which can lead to end organ damage if not promptly diagnosed and treated. Treatment options include medical management with antihypertensive agents, balloon or stent angioplasties, surgical revascularization, and nephrectomy. The aim of the study was to review the efficacy of antihypertensive therapy only in the management of FMD in a very young child. **Methods**: This is a retrospective chart study with review of literature. **Results**: Here, we report a 22-month-old toddler who presented with severe resistant hypertension and cardiomyopathy who was found to have focal FMD of the right renal artery. She also presented with proteinuria, hyponatremia that was probably secondary to pressure natriuresis, hypokalemia, hyperaldosteronism, and elevated plasma renin activity. The stabilization of blood pressures was done medically with the usage of antihypertensive medications only, without the need for angioplasty or surgical revascularization. **Conclusions**: We demonstrate that surgical intervention may not always be necessary in the treatment of all cases of FMD, especially in a small child where such intervention may be technically challenging and lead to potential complications. Hence, medical management alone may be sufficient, at least for the short-term, in small children with controlled hypertension and normal renal function, with surgical intervention reserved for FMD with medication-refractory hypertension and/or compromised renal function.

## 1. Introduction

Renal artery stenosis (RAS) is a potentially treatable but under-recognized cause of hypertension in children. Timely diagnosis is of utmost importance to plan appropriate therapeutic intervention. Some studies have shown that RAS is the underlying etiology in 8%–10% of all cases of pediatric hypertension [1,2]. Fibromuscular dysplasia (FMD) accounts for about one third of all cases of pediatric RAS [1]. FMD can occur in non-syndromic children, including Takayasu’s arteritis [3] as well as in children with Williams’ syndrome, neurofibromatosis, Marfan syndrome, and Ehlers–Danlos syndrome, among others [2,4]. FMD is an idiopathic, segmental, non-inflammatory, and non-atherosclerotic disorder that causes arterial stenosis with potential complications including aneurysm, dissection and tortuosity [5]. The “string-of-beads” is a classic appearance, but other manifestations such as vascular loops and fusiform vascular ectasia are not uncommon. Renal arteries (RAs) are involved in up to 75% of patients with FMD [6,7]; other common vessels that are involved are the extra-cranial carotid and vertebral arteries [7]. However, nearly all arterial beds may be affected, and multi-vessel involvement is common [6]. FMD may be familial in about 10% of cases; autosomal dominant patterns with incomplete penetrance and variable clinical manifestations have been shown [8]. The majority of adult FMD patients are female, and the right kidney seems to be affected more than the left [7]. Among children, the data from the United States Registry for FMD have shown that the mean age at diagnosis is 8.4 ± 4.8 years, with a range of age at diagnosis of 16 days to 17 years [9]. Additionally, the gender difference (female to male ratio) is much less prominent in children (3:2 compared to 47:3 in adults) [6,9]. Though any vasculature can be affected, the RAs are involved in 97% of children with FMD; mesenteric and abdominal aorta are other commonly affected sites, with carotids less affected in children than in adults [9,10]. A renal bruit is heard in only a very small subset of children with FMD [11]. The management of renal artery FMD includes medical therapy and surveillance, endovascular therapy (angioplasty with or without stenting), or surgery. The purpose of this case study was to demonstrate the therapeutic efficacy of antihypertensive agents alone in the management of unilateral FMD of the RA, without the need for endovascular or surgical revascularization. A 22-month-old toddler with focal FMD of the right RA whose hypertension was controlled only with antihypertensive agents is described in this study. 

## 2. Materials and Methods

This is a retrospective chart study with a review of the literature, and the family of patient gave informed consent for inclusion in this study. 

A renal bladder sonogram was performed with Doppler images. A duplex study of the renal artery was performed at a vascular lab. The velocities were measured in cm/s, the diameters were measured in mm, and the pressures were measured in mmHg. Peak systolic velocities were calculated at the levels of the aorta, the renal hilum, and different segments of the renal artery (proximal, mid and distal). Resistive indices were calculated at the levels of the main, hilar and interlobar renal arteries. A complete 2-D, M-mode, color, and a spectral Doppler echocardiographic study was performed. Computed tomography angiograms of the head and neck vessels, the chest, the abdomen, and the pelvis were performed with intravenous contrast. Radionuclide renography was performed with the administration of a dose of 3.3 mCi of Tc-99m mercaptoacetyltriglycine-3, and a posterior flow study of the abdomen was performed at 2 sec/frame. Then, a standard renogram was continued in the posterior projection, with imaging at 1 minute per frame for 27 minutes. Images were summed to create a series of 3-minute sequential images. At 10 minutes into the study, 40 mg of furosemide was intravenously administered. Background-corrected time–activity curves were generated with calculations of time-to-peak activity and half-times of clearance. Next, to evaluate the effect of gravity on the drainage from the upper collecting systems, a delayed image was obtained at 15 minutes after the standard renogram. A digital subtraction angiogram was performed at the interventional radiology laboratory with fluoroscopic guidance. The patient was placed supine on the angiography table. Both groins were prepped and draped in normal sterile fashion. An appropriate arterial access site was fluoroscopically and sonographically determined. A static ultrasound image was obtained. Under real-time sonogram guidance, a 21-gauge needle was advanced into the femoral artery. A microwire was advanced through the needle to secure vascular access. The needle was exchanged for a 3/4 micropuncture catheter. The wire and inner catheter were removed and exchanged for a 0.025-inch Torx wire, which was advanced into the aorta. The outer catheter was removed and exchanged for a 3 French Junior SOS catheter, which was advanced into and formed in the aorta. Selective right and left renal artery angiographies were performed. At the end of the procedure, the catheter was unformed in the aorta and removed. Hemostasis was achieved with manual compression. All wire and catheter manipulations were performed under fluoroscopic guidance.

## 3. Case Presentation

A 22-month-old African-American girl was transferred from an outside hospital to the intensive care unit for further evaluation and the management of uncontrolled severe hypertension. She had a three-day history of fever, cough, rhinorrhea, shortness of breath, dizziness and vomiting. She was born at term with a weight of 2.8 kg and had an unremarkable perinatal course. She had no known genetic syndromes. Besides a history of difficulty gaining weight with a current weight of 10.7 kg (seventh percentile) and a height of the 25th centile, she had no other known medical problems in the past. There was no history of urinary tract infections or blood in urine. She did not have headaches, seizures or fainting episodes. Immunizations were up-to-date. Prior blood pressure (BP) measurements were unknown because they were never obtained. There were no other past hospitalizations. She had not been taking any medications. Family history was unremarkable for kidney disease, hypertension, dialysis or transplantation. 

Upon physical examination, her vital signs were as follows: a temperature of 36.9 °C (98 °F), a respiratory rate of 16 per minute, a heart rate of 145 beats per minute, an oxygen saturation of 100% on room air, and an elevated BP of 200/120 mm Hg (manual, right upper extremity, proper size cuff; 95th percentile value for her age, gender and height 104/62 mm Hg). The examination showed a thin-appearing toddler who was not in obvious distress. The rest of the examination was negative except for tachycardia and a grade II/VI systolic murmur at the apex. A renal bruit was not heard upon auscultation. There was no skin rash or edema. The neurologic examination was normal. 

The nasal swab for a respiratory viral panel test was positive for parainfluenza 3. Besides mild nasal congestion, she had no cough, difficulty breathing, or wheezing, and she remained stable in room air. She remained afebrile throughout the hospital admission. Her throat culture was negative for beta-hemolytic streptococcus. Her blood culture showed no growth. Her chest x-ray showed mild cardiomegaly but no focal airspace disease, pneumothorax or pleural effusion. Her electrocardiogram showed supraventricular tachycardia. Her echocardiogram revealed a dilated left ventricle with severe concentric left ventricular (LV) hypertrophy, with an LV mass index of 187 gm/m^2.7^ and a mildly depressed LV systolic function with an ejection fraction of 50%. There was a thrombus in the apex of the LV. The serum brain natriuretic peptide and troponin I levels were 1565 pg/mL (normal: <100 pg/mL) and 242 pg/mL (normal: <15 pg/mL), respectively. For possible viral-induced cardiomyopathy, she was started on intravenous immunoglobulin and methylprednisolone. For hypertension, she was started on nicardipine infusion, along with nitroprusside with a goal BP reduction of 30% over 24 hours. Milrinone was added for depressed LV systolic function. She remained on the maximum nicardipine infusion rate for several days due to persistent hypertension, with systolic BPs of 140–150 mm Hg. Heparin was transitioned to enoxaparin and aspirin, with the resolution of the cardiac thrombus by day three. A non-contrast computed tomography (CT) scan of head showed no mass, acute intracranial hemorrhage, cytotoxic edema, midline shift, or signs of herniation. Her urine output was normal. 

A renal function test showed an initial serum sodium level of 131 meq/L with a nadir of 121 meq/L at day three of admission, a serum chloride level of 97 meq/L (nadir 84 meq/L at day three), a serum potassium level of 3.2 meq/L, a blood urea nitrogen level of 25 mg/dL, and a serum creatinine level of 0.5 mg/dL. Her fasting blood glucose level was 90 mg/dL. Her urine sodium level was 89 mmol/L with a fractional excretion of sodium of 1.5%. Her serum albumin level was between 2.1 and 2.5 gm/dL, and her random urine protein to creatinine ratio was 4.6 (normal: <0.2). There was no microscopic hematuria. Her urine culture showed no growth. Her blood counts and serum complements were both normal. Her sickle cell screen was negative. A renal sonogram showed a small right kidney of 6 cm in length (36th centile) and a left kidney of 7.7 cm in length (96th centile), with good blood flow in both kidneys. Her serum aldosterone and plasma renin activity at the time of admission were both elevated at 106 ng/dL (normal: <16 ng/dL, supine, quantitative chemiluminescent immunoassay) and >150 ng/mL/h, respectively (normal: 1.7–11.2 ng/mL/h, supine, quantitative enzyme-linked immunosorbent assay; both ARUP laboratories, Salt Lake City, UT, USA). 

A work-up for possible catecholamine-secreting tumors due to severe hypertension included the evaluation of urine and serum catecholamines. Her 24-hour urine normetanephrine, metanephrine, vanillylmandelic acid, and homovanillic acid levels were 1525 µg/g of creatinine (normal: 0–1300 µg/g of creatinine), 433 µg/g of creatinine (normal: 0–530 µg/g of creatinine), 10 mg/g of creatinine (normal: <16 mg/g of creatinine), and 12 mg/g of creatinine (normal: <25 mg/g of creatinine), respectively. Her serum fractionated catecholamines showed the following: an epinephrine level of 152 pg/mL (normal: 36–640 pg/mL) and a norepinephrine level of 1964 pg/mL (normal: 68–1810 pg/ml), respectively. Given a less than two-fold increase from normal range of serum norepinephrine, the suspicion of the presence of a neuroendocrine tumor was low. 

A diuretic nuclear renogram showed asymmetrical renal function with decreased blood flow and function in the small right kidney relative to the left. Her differential function was 70% in the left and 30% in the right. There was a normal uptake of the technetium-99 m MAG3 (mercaptoacetyltriglycine) with prompt excretion from the left kidney. Her right kidney had little delayed uptake with prompt excretion (Figure 1 and Figure 2, Table 1). A duplex sonogram of her RA showed focal elevation in velocity in the proximal right main RA (peak systolic velocity (PSV): 405 cm/second) with a decreased velocity in the mid and distal RAs (PSV: 117 and 110 cm/sec, respectively). The renal aortic ratio (RAR, the ratio of renal to aortic PSV) was elevated at 3.2. Additionally, distinct “tardus-parvus” (tardus-slow and late, parvus-small and little) waveforms were seen at the right hilar and interlobar arteries. This combination of an elevated PSV at the proximal right main RA along with an elevated RAR (usual cut-off 3) and abnormal waveforms was highly suggestive of proximal right main RAS. Velocities in the left main RA were normal with no differences at various segments of the main artery. A CT angiogram (CTA) of the RAs showed a small right kidney of 6 cm in length with a decreased enhancement of the superior pole of the right kidney. The origin of the right main RA was diminutive compared to the left. There was a homogenous enhancement of the left kidney with a normal appearing left RA. Hypertension at that time was well controlled with amlodipine, clonidine, furosemide, spironolactone and labetalol. Nicardipine infusion was discontinued at day seven of admission. 

A few days later, she underwent a digital subtraction angiogram (DSA), which showed a beaded appearance of the right proximal main RA, which caused a focal stenosis with post-stenotic dilatation (Figure 3 and Figure 4). The pressure gradient across the stenotic lesion was 40 mm Hg. There was normal renal parenchymal perfusion. The left RA demonstrated a normal appearance of the main and branch arteries with normal renal parenchymal perfusion and no evidence of stenosis and irregularity. A diagnosis of right RA stenosis, most likely due to focal FMD, was made. In conjunction with vascular surgery and intervention radiology, it was collectively decided not to pursue the intervention of the stenotic lesion due to the risk of complication, specifically the injury or dissection of the right RA, especially because her BPs were relatively controlled on five antihypertensive agents. Subsequent to the angiogram, enalapril was added once bilateral RA stenosis was excluded for optimal BP control. A CTA of the head and neck vessels to look for evidence of FMD in the head and neck showed a normal vasculature of the vessels. Baby aspirin was started to prevent thromboembolic events. 

Though fever, parainfluenza virus infection, and LV thrombus could have partly contributed to the elevated BPs initially, these signs and symptoms had resolved at the time of or soon after admission. Additionally, the finding of a discrete anatomic lesion in the right RA that is known to cause hypertension suggests that the right RA FMD indeed was the most probable cause of the severe, long-standing hypertension, as suggested by the severe concentric LV hypertrophy. At the time of discharge two weeks later, her BPs were relatively optimal (105–110 mm Hg SBPs) on enalapril, amlodipine and labetalol. Subsequent monthly clinic follow-up visits seven months later have shown stable SBPs with no clinical symptoms of hypertension. Her renal function has remained stable with normal serum electrolytes without hyponatremia and hypokalemia. Her proteinuria has resolved. A follow-up echocardiogram showed normal LV systolic function and stable but persistent LV hypertrophy with no reappearance of the thrombus that was seen in the first echocardiogram. For now, we plan to closely follow-her BPs, along with a follow-up diuretic renogram and a DSA in a few months, with a tentative plan for either continued medical treatment or surgical intervention (angioplasty with or without stent and surgical revascularization or right nephrectomy) based on the clinical course. 

## 4. Discussion

Renal FMD should be suspected in a patient with malignant and resistant hypertension, a unilateral small kidney of unclear etiology, the presence of an abdominal bruit, and known FMD in another vascular territory. Our patient was a young female toddler who had resistant malignant hypertension and had a small right kidney but did not have FMD at other vascular sites. Since obtaining the histology of the affected RA is an invasive procedure and is hence not always feasible, a binary angiographic classification of FMD was devised: focal versus multifocal FMD [12]. Multifocal FMD is more common than focal FMD (82% vs 18%) in adults [12], and it mainly occurs in the middle and distal RAs as opposed to focal FMD, which has no site predilection [13]. Additionally, focal FMD is more common in children [9]. Intimal fibroplasia is the most common type of FMD in the focal type (vs medial fibroplasia in the multifocal type) and presents as a distinct focal stenosis [14]. Our patient presented with the focal type of FMD with post-stenotic dilatation. 

Most centers perform a Doppler sonogram as an initial screening test in suspected RAS and CTA, or they perform magnetic resonance angiogram (MRA) as the second step-procedure. One study looked at 58 patients with suspected RAS and found the following sensitivity, specificity, and positive and negative predictive values (PPV and NPV) for Doppler sonograms, CTAs and MRAs: 75%, 89.6%, 60% and 94.6%; 94%, 93%, 71%, and 99%; and 90%, 94.1%, 75%, and 98%, respectively [15]. Meyers et al. in their single center study of 25 children with FMD showed CTA to be the most sensitive among the three modalities, but the study also produced 30% false positive results. The DSA was the only procedure that was 100% sensitive and specific, had 100% PPV and NPV, and the showed the distal vessel disease [11]. Recently, the European Society of Hypertension and Society for Vascular Medicine’s first international Consensus on FMD also recommended the CTA of the RAs as the first non-invasive imaging modality of choice [5], with an RA duplex reserved only for centers with extensive experience with the duplex. A dedicated RA duplex is a valuable study modality in screening RAS because it provides valuable information about the approximate site and severity of stenosis by measuring the PSV, the RAR, the types of waveforms, the end-diastolic velocity (EDV), and resistive indices ((PSV-EDV)/PSV) at different segments of the RA, as was done in our patient [16,17]. A PSV greater than 200 cm/sec has been suggested as the threshold for the Doppler diagnosis of a 60% or more reduction of RA diameter (sensitivity 97%, specificity 72%, PPV 81%, and NPV 95% in terms of diagnostic accuracy for RA stenosis) [17]. Though earlier studies have shown some efficacy of captopril-enhanced 99Tcm-MAG3 renal scintigraphy in RAS, lately studies of this nature have has fallen out of favor and are not recommended by the FMD consensus committee [5,18,19]. However, the MAG-3 renal scan may have a role in patients with uncontrolled RAS despite medical management or revascularization to see whether nephrectomy is an option by measuring the function of the affected kidney. In our patient, the affected right kidney had a function of 30%, and, since her BPs were controlled on medical therapy, we did not perform nephrectomy. DSA remains the gold standard modality due to its diagnostic and therapeutic advantages. 

Therapeutic decision in RAS, including renal artery FMD, depends on the nature and location of vascular lesion (stenosis versus dissection versus aneurysm), the severity of hypertension and efficacy of the anti-hypertensive therapy, the size of the patient, and comorbid conditions. Meta-analyses of randomized trials and Cochrane reviews have shown that there is not an enough evidence to support the routine use of revascularization procedures (ballooning and/or stenting) in the treatment of symptomatic patients with RAS [20,21,22]. A 2003 Cochrane systematic review of three randomized controlled trials involving 210 patients with RAS was unable to conclude that balloon angioplasty is superior to medical therapy in lowering BP in those in whom BP can be controlled with medical therapy [20]. However, the study did find that in patients with refractory hypertension, balloon angioplasty more effectively lowered BP than medical therapy alone. Similarly, a subsequent Cochrane review published in 2014 showed insufficient evidence that revascularization with or without stenting is superior to medical therapy in hypertensive patients with RAS [21]. It is important to mention that both of these reviews looked at adult patients with RAS secondary to atherosclerosis. In regard to RAS that is secondary to FMD in adults, one meta-analysis of 11 clinical studies showed a hypertension cure rate of 36% (defined as BP < 140/90 mm Hg). The success rate was higher in younger patient group at the time of treatment and those with shorter duration of hypertension. Repeated angioplasties were needed in 18% of patients [23]. Similarly, some single center studies with long-term follow-ups have reported significantly favorable short and long-term clinical outcomes with revascularization, in adults with FMD [24].

In children, a retrospective single center study in 12 children with RAS who underwent percutaneous balloon and stent angioplasties showed that only stent angioplasty was able to significantly reduce the BP. The study also found a restenosis rate of 40% with balloon angioplasty alone and no stenosis in the stent angioplasty group [25]. Zhu et al. reported 34 cases of RAS in children who underwent percutaneous transluminal angioplasty (PTLA) with a success rate of 94%; the restenosis rate was 20% (within 3–47 months post angioplasty) without stenting, and 27% of the patients failed to respond to angioplasty [26]. Konig et al. summarized the efficacy of balloon angioplasty by studying 68 studies in children, and they ultimately showed variable success but no clear outcome with stent angioplasty [27]. Shroff et al., in their 20-year experience from a single center, showed a restenosis rates of 37% following stenting procedures [28]. The localization of the stenotic lesion in the renal artery also seems to influence the outcome of PTLA: An isolated lesion in the mid or distal part of the artery may have a better outcome than lesions near the origin of the artery or long or multiple stenosis [29]. Data from the pediatric renal FMD registry, the largest cohort of FMD children reported to date, have shown that at the time of enrollment, 88% of children have been medically managed with antihypertensive agents alone with a median number of agents of 1. Subsequently, 54% of these children have required surgical intervention, mostly in the form of balloon angioplasty [9]. This data showed that about half of the FMD children responded to antihypertensive agent only. Additionally, in those who require endovascular intervention, the consensus committee on FMD recommends angioplasty alone as the approach of choice for renal artery FMD, with stenting reserved only for procedural complications such as arterial dissection or rupture. However, in very young children with small renal vessels, angioplasty or surgical repair is technically difficult and may not be successful [30]. Thus, long–term antihypertensive therapy may be the only option until vessel size is larger as long as the BPs remain well controlled, as in our patient. Nephrectomy may be required even in small children if hypertension is difficult to control, especially if the affected kidney’s function is also poor [31]. Hence, a close follow-up is necessary to determine appropriate clinical management. 

Some other clinical associations with RAS that occurred in this patient and are worth mentioning are the nephrotic range proteinuria and hyponatremic hypertensive syndrome, both of which have been described in the literature [31,32]. The histology of the kidney in patients with RAS, nephrotic syndrome, and hyperreninemia who underwent nephrectomy for uncontrolled hypertension have shown severe foot process effacement [33]. In unilateral RAS, renin that is released by the affected ischemic kidney causes an increase in circulating angiotensin II, thereby constricting the efferent arteriole of the contralateral kidney. This subsequently results in an increase in the hydrostatic pressure in the glomerulus, causing an increase in the filtration fraction, as well as an increase in glomerular capillary permeability, thereby leading to nephrotic-range proteinuria [34]. One study found that when renal angioplasty led to the curing of hypertension, it also resulted in the significant reduction of proteinuria; however, no change in proteinuria occurred if the BP did not decrease [35]. The hyponatremia in RAS is most likely secondary to pressure natriuresis, which is a central component of the feedback system for the long-term control of arterial pressure [36]. The increase in renal perfusion pressure from renin-induced hypertension leads to a decrease in tubular sodium reabsorption and, hence, an increase in sodium excretion. This has been shown to be secondary to an increase in medullary blood flow and renal interstitial hydrostatic pressure, as well as renal autacoids such as nitric oxide, prostaglandins, kinins; it has also been shown to be secondary to a decrease in angiotensin II. Both nephrotic proteinuria and hyponatremia resolved in our patient with adequate control of BPs. 

## 5. Conclusions

Surgical intervention in renal FMD poses a challenge in small children, and, hence, treatment options become limited. The question of whether to only manage medically or by endovascular or surgical intervention in someone with narrow vessels and the associated potential risks is a challenging one that must be considered on a case by case basis depending on the clinical profile of each patient. A very close long-term follow-up of blood pressures, renal function, the involvement of other vasculatures, and signs of end organ damage from chronic hypertension is required in these patients. Even if short term medical management with antihypertensive therapy only may be sufficient in very young children, endovascular or surgical revascularization and/or nephrectomy may be required later with progressive refractory hypertension and/or compromised renal function. Longitudinal prospective studies of children who have been managed with medical management alone are needed to assess the efficacy of such intervention.

## Figures and Tables

**Figure 1 medicines-07-00009-f001:**
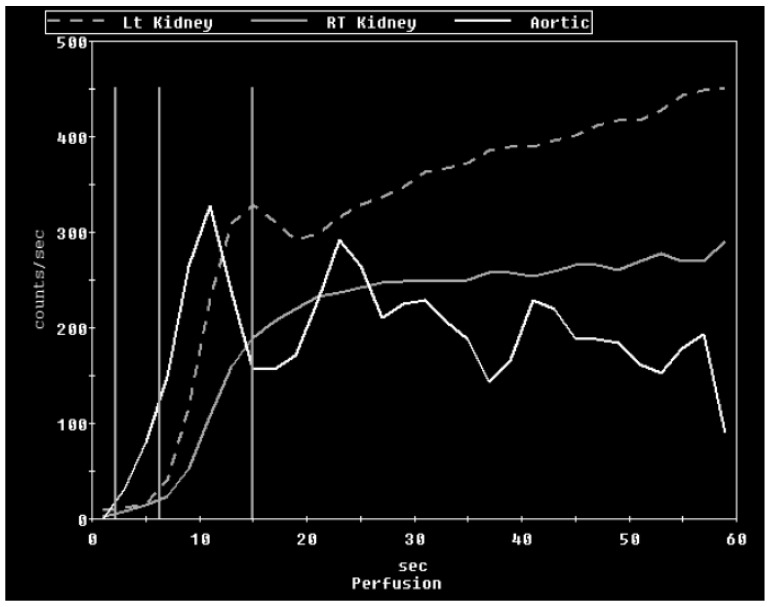
MAG-3 diuretic renal scan showing the time–activity curves in the perfusion phase. The right kidney had a slightly delayed uptake of MAG-3 with a reduced flow. The x-axis is labelled as radiotracer activity in counts/sec, and the y-axis is labelled as time in seconds following the infusion of the radiotracer.

**Figure 2 medicines-07-00009-f002:**
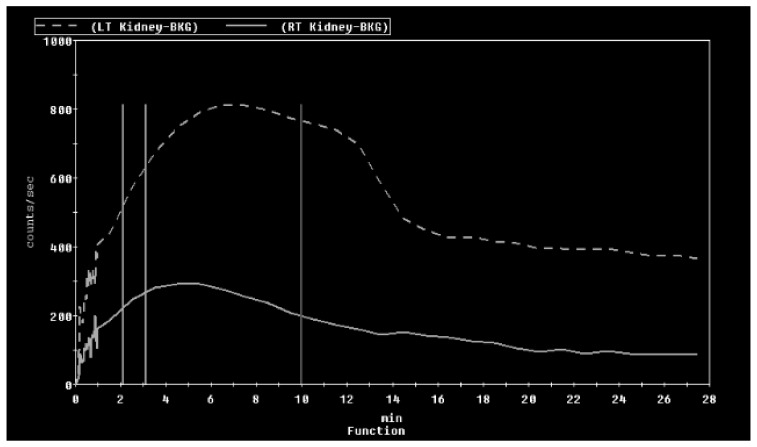
MAG-3 diuretic renal scan showing the time–activity curves in the drainage phase. MAG-3 was promptly excreted from the left kidney, which was augmented by furosemide. MAG-3 was progressively drained from the right kidney with no acceleration with furosemide. The x-axis is labelled as radiotracer activity in counts/sec, and the y-axis is labelled as time in minutes following the infusion of the radiotracer.

**Figure 3 medicines-07-00009-f003:**
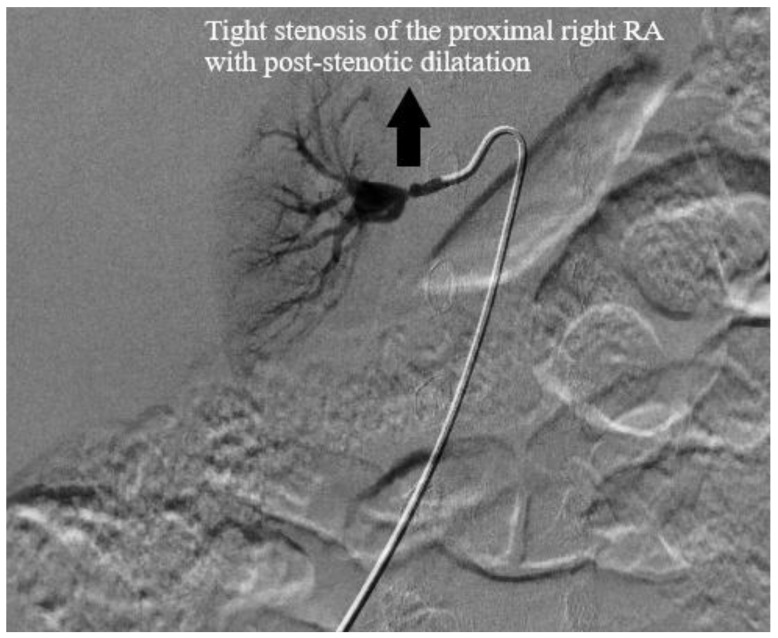
Digital subtraction angiography of the right renal artery, which showed proximal renal artery stenosis.

**Figure 4 medicines-07-00009-f004:**
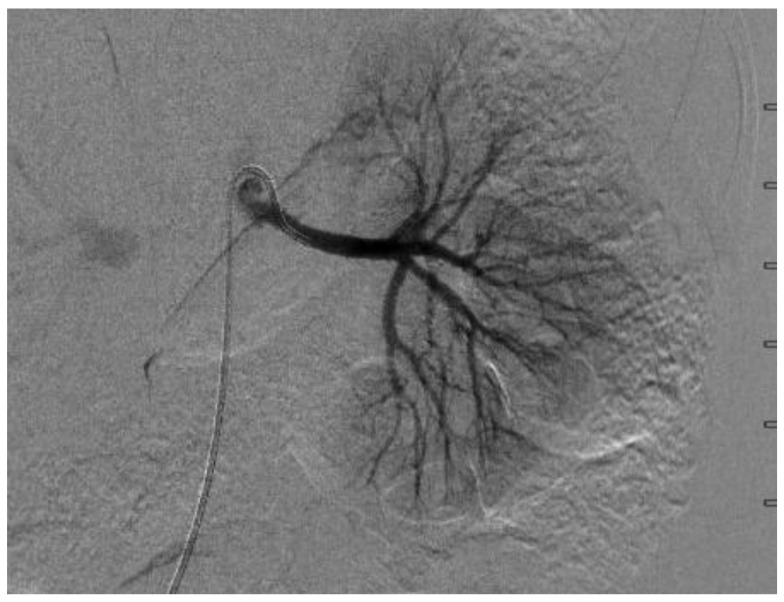
Digital subtraction angiography of the left renal artery, which showed no stenosis.

**Table 1 medicines-07-00009-t001:** MAG-3 (mercaptoacetyltriglycine) diuretic renal scan analysis panel that shows the differential renal function and diuretic T_1/2_ time.

Kidney	Left	Right
Kidney Area (cm^2^)	37.39	29.67
Perfusion % (Int)	66.25	33.75
Perfusion % (Slo)	81.11	18.89
Uptake % (Int)	70.13	29.87
Time to peak (min)	6.39	5.39
Diuretic T_1/2_ (min)	15.5	10.5

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
