# Peer review of "Efficacy of Antihypertensive Therapy in a Child with Unilateral Focal Fibromuscular Dysplasia of the Renal Artery: A Case Study and Review of Literature"

_medicines, 2020, doi:10.3390/medicines7020009_

Round 1

Reviewer 1 Report

The authors presented the case of FMD in toddler showing the hypertension probably due to the right renal arterial stenosis due to FMD. They demonstrated that the anti-hypertensive therapy was effective, with no IVR or surgical treatment.

The focus of this report might be important however, as the current form of the manuscript may mislead the therapeutic decision making of severe FMD patients. Therefore, authors need to consider the following points before the further consideration of publication of this manuscript.

There are not a few cases of FMD and rescued by the surgical or IVR treatment. The authors may want to show such cases, not just strongly focusing on the anti-HT medication. Actually, in most of the cases, such kind of medications are used initially and then for the refractory cases, further interventional therapies have been conducted. As there are several vessels might be involved in FMD, the authors need to show whether this case had other FMD lesions. (for e.g. in the cerebral artery) As this case showed severe fever, infection, and thrombus in the heart, these conditions also contributed to the increase of blood pressure. The authors also need to describe the detail clinical course of these conditions and discuss regarding these factors which could contribute to increase the BP.

Author Response

Reviewer 1

Thank you for your valuable suggestions. 

  1. We have modified and added statements in the abstract as per your suggestion. "We demonstrate that surgical intervention may not always be necessary in the treatment of all cases of FMD, especially in a small child where such intervention may be technically challenging leading to potential complications. Hence, medical management may be sufficient, at least for a short -term, in small children with controlled hypertension and normal renal function, with surgical intervention reserved for FMD with medication-refractory hypertension and/or compromised renal function".
  2. As per your suggestion, we have added statements in the discussion section about the benefits of endovascular and/or surgical revascularization in children with severe FMD or medication refractory FMD, and not just focusing on the utility of antihypertensive medications (Discussion-paragraph 4, reference 9). 
  3. We have mentioned that we also did the CT angiogram of head, neck, chest with intravenous contrast to look at evidence of FMD at other areas of the body (Case presentation- 7th paragraph- second last line).
  4.  As per suggestion, we have described the detail clinical course of fever, infection and thrombus and added statement that these could have contributed to the hypertension as well (Case presentation- 3rd paragraph- 2nd statement; 8th paragraph- 1st and 2nd statements)

Thank you for the recommendations. 

Reviewer 2 Report

The paper titled  Efficacy of antihypertensive therapy in unilateral focal fibromuscular dysplasia of the renal artery, presents a case study of  22-month-old African-American girl with severe hypertension.

Although the paper is interesting, I have some major concerns:

Title

As the paper presents a case study it should be included in the paper title.

Abstract

The abstract is lacking informative conclusion. It should be written in more details. Moreover, the aim of the study should be included in the abstract.

Introduction

Authors did not included the aim of the study. It should be clearly stated what is the purpose of this study.

Material and Methods

A part describing analysed patient should be included in the Material section.

Technical description of applied devices should be included in the Material section.

Results

Image 1c should be presented as a table not a figure.

Conclusions

Conclusions should reflect the most important findings.

Figures

Image 1A and Image 1B are unreadable. The description of both axis should be enlarged to be readable. Moreover, units should be included.

Image 1B should be anonymised.

Author Response

Thank you for the valuable suggestions. 

  1. We have modified the title to include the case study. 
  2. We have modified the abstract as per your suggestion to include more details. Aim of the study has been added just before the material section in the abstract
  3. We have added the aim/purpose of the study in the introduction. 
  4. We have added the detailed technical description of the studies that were performed in this patient in the Material section. 
  5. Image 1C has now been presented as a table. 
  6. Conclusion has been modified to include the most important findings as suggested. 
  7.  New images of 1A and 1B have been resubmitted as best as we could. We have described the labels of both axis in the legend. Units have been described in the legend. Image 1B has been anonymized. 

Thank you for the recommendations. 

Round 2

Reviewer 1 Report

The paper has been modified and can be acceptable as a case report.

Reviewer 2 Report

None